The extended impact of the COVID-19 pandemic on medical imaging case volumes: a retrospective study

Alhazmi Fahad H. fhdhazmi@taibahu.edu.sa 1
Alrehily Faisal A. 1
Alsharif Walaa 1
Gameraddin Moawia 1 2
Alsultan Kamal D. 1
Alsaedi Hassan Ibrahim 3
Aloufi Khalid M. 1
Alshoabi Sultan Abdulwadoud 1
Abdulaal Osamah M. 1
Qurashi Abdulaziz A. 1
1 Department of Diagnostic Radiology, College of Applied Medical Sciences, Taibah University , Almadinah Almunawarah , Saudi Arabia
2 Department of Diagnostic Radiology, Faculty of Radiological Sciences and Medical Imaging, Alzaiem Alzhari University , Khartoum , Sudan
3 Medical Imaging Department, King Abdulaziz Medical City , Jeddah , Saudi Arabia
Fan Chengming
Electronic publication date: 2025 Mar 5
Publication date: 2025
Volume: 13
Electronic Location ID: e18987
Received 2024 Oct 15; Accepted 2025 Jan 22
Copyright: ©2025 Alhazmi et al.
Copyright year: 2025
Copyright holder: Alhazmi et al.
License: This is an open access article distributed under the terms of the Creative Commons Attribution License, which permits unrestricted use, distribution, reproduction and adaptation in any medium and for any purpose provided that it is properly attributed. For attribution, the original author(s), title, publication source (PeerJ) and either DOI or URL of the article must be cited.
License URL: https://creativecommons.org/licenses/by/4.0/

Keywords: Radiology, Medical imaging, COVID-19, Pandemic

Funding: The authors received no funding for this work.

==============================
Objective

This study aims to investigate the long-term effects of the COVID-19 pandemic on medical imaging case volumes.

Methods

This retrospective study analyzed data from the Integrated Radiology Information System-Picture Archive and Communication System (RIS-PACS), including monthly medical imaging case volumes at a public hospital, spanning from January 2019 to December 2022. The study collected data on medical imaging examinations, comparing the pre COVID-19 period, which acted as a control group, with the periods following COVID-19, which were designated as cohort groups.

Results

The total number of medical imaging procedures performed (n = 597,645) was found significantly different (F = 6.69, P < 0.001) between 2019 and 2022. Specifically, the bone mineral density/computed radiography (BMD/CR) modality experienced a significant decrease (P = 0.01) of the procedures performed in 2020 and 2021 compared to 2019. Conversely, the nuclear medicine/computed tomography (NM/CT) and computed tomography (CT) modalities demonstrated a significant increase of the procedures performed in 2021 (P = 0.04) and (P < 0.0001), respectively, and in 2022 (P = 0.0095) and (P < 0.0001), respectively, compared to the pre-pandemic year. The digital X-ray modality (DX) showed the highest volume (67.63%) of the performed procedures overall between 2019 and 2022. Meanwhile, magnetic resonance imaging (MR) and ultrasound (US) modalities experienced a slight drop in the number of procedures in 2020—4.47% for MR and 1.00% for US, which subsequently recovered by 22.15% and 19.74% in 2021, and 24.36% and 17.40% in 2022, respectively, compared to 2019.

Conclusion

The COVID-19 pandemic initially led to a drop in the number of medical imaging procedures performed in 2020, with the most noticeable drop occurring during the early waves of the pandemic. However, this trend revealed a gradual recovery in the subsequent years, 2021 and 2022, as healthcare systems adapted, and pandemic-related restrictions were modified.

Introduction

From 2019 to 2022, the COVID-19 pandemic, caused by the SARS-CoV-2 virus, dramatically reshaped global healthcare systems, which accelerated the adoption of telehealth, enabling remote patient care. In December 2019, pneumonia cases of unknown origin were reported in Wuhan, China. By January 2020, the virus began spreading beyond China, with the first case reported in the United States. By early 2021, vaccines like Pfizer-BioNTech and Moderna were authorized for emergency use. The emergence of variants, such as Delta, led to renewed restrictions in some regions around the world. In 2022, countries began easing measures and shifted focus to managing COVID-19 as an endemic virus, emphasizing the need for global cooperation, scientific advancements in public health, and addressing misinformation.

Since the World Health Organization (WHO) declared COVID-19 a global pandemic in 2020 and subsequently downgraded it from a global health emergency in 2023, the healthcare landscape has experienced significant changes. The rapid spread of the virus led to an escalation in morbidity and mortality rates (Rajpal, Rahimi & Ismail-Beigi, 2020; Metz et al., 2022; Chang et al., 2022), fundamentally altering patient admissions and visitation patterns (Birkmeyer et al., 2020; Rennert-May et al., 2021), and necessitating the postponement of numerous elective procedures (Kuitunen et al., 2021; Pyrgidis, Sokolakis & Hatzichristodoulou, 2022; Werger et al., 2022; Hunger et al., 2022; Frio et al., 2022). This pandemic has notably reshaped both inpatient and outpatient healthcare services in terms of to be more flexible and adapt more easily to exceptional event is required to prevent future crisis (Jordan-Rios et al., 2023).

Particularly in the field of radiology, the pandemic’s impact has been multifaceted and profound. Economic challenges have emerged (Cavallo & Forman, 2020; Alhazmi et al., 2024), alongside shifts in workload distribution and management (Masood et al., 2022; Alhazmi et al., 2024). The scheduling and conduct of radiology appointments have undergone significant changes (Langan et al., 2021; Alhazmi et al., 2024), necessitating adaptations in patient care and workplace safety protocols (Salih et al., 2022). These shifts have had far-reaching implications for the profession at large (Coppola et al., 2021), affecting radiology trainees and educational landscape (Alvin et al., 2020; Majumder et al., 2021), influencing research directions and priorities (Vagal et al., 2020), and impacting the personal wellness of radiographers (Flood, McFadden & Shepherd, 2022). The resilience demonstrated by medical staff in the radiology departments of Saudi Arabia during the COVID-19 pandemic was found at a moderate or intermediate level, which suggests that healthcare administrations should develop strategies to better equip these workers in effectively coping with the adversities they face in the workplace (Asiri et al., 2023). Additionally, an assessment conducted in Al-Qassim, Saudi Arabia, revealed that the COVID-19 pandemic has had a negative impact on radiology department employees and trainees, adversely affecting training and medical education (Albweady et al., 2024).

Regarding medical imaging case volumes, there has been a significant reduction due to COVID-19 pandemic in countries such as Saudi Arabia (The Saudi Food & Drug Authority (SFDA), 2021; Alelyani et al., 2021), United Arab Emirates (Salih et al., 2022), Qatar (Al Kuwari et al., 2022), Jordan (Gharaibeh et al., 2023), the United States (Naidich et al., 2020), the United Kingdom (Pinson et al., 2022), Germany (Schmidbauer et al., 2022) and Australia (Rizzetto et al., 2023). Given the extensive impact of the pandemic, a comprehensive investigation into the long-term effects of the pandemic on medical imaging case volumes and modalities is necessary.

The study aims to assess the impact of the COVID-19 pandemic on medical imaging volumes across various modalities. It seeks to determine which medical imaging modalities were affected by the restrictions imposed due to COVID-19 and how the volume of medical imaging changed throughout the pandemic. This would help to identify best practices and inform future healthcare policy and preparedness strategies, with the goal of maintaining essential medical imaging services during health crises and ensuring the resilience and responsiveness of healthcare delivery systems. To the best of our knowledge, this is the first study that has investigated the extended impact of the COVID-19 pandemic on medical imaging case volumes in the region.

Materials & Methods

Ethical considerations

This study has obtained ethical approval from the Institutional Review Board of the General Directorate of Health Affairs in Almadinah Almunawarah, Saudi Arabia (Ref. 031-22).

Study settings

This is a single-site retrospective study analyzing data obtained from the Integrated Radiology Information System–Picture Archive and Communication System (RIS-PACS) at a public hospital in Almadinah Almunawarah, Saudi Arabia. This site was selected because it is regarded as the largest hospital in the region, with a capacity of 500 beds. It has served a total of 266,992 patients, including 138,921 emergency cases and 128,001 outpatients across all medical specialties.

The study encompasses monthly medical imaging case volumes from January 2019 to December 2022. Ten affiliated imaging modalities were included: bone mineral density/computed radiography (BMD/CR), computed tomography (CT), digital X-ray (DX), mammography (MG), magnetic resonance imaging (MR), nuclear medicine (NM), nuclear medicine/computed tomography (NM/CT), radiographic fluoroscopy (RF), ultrasound (US) and X-ray angiography (XA).

Data collection

An aggregate number of medical imaging examinations was collected, comparing the pre COVID-19 period (serving as the control group) with the post COVID-19 periods (serving as the cohort groups). The study focused on monthly medical imaging case volume, including inpatient, emergency, and outpatient examinations. The total volume of medical imaging case volume was analyzed to assess the changes occurring during the period. Additionally, variations in the annual and monthly number of procedures performed across each imaging modality were calculated.

Data analysis

For the descriptive analysis of changes in medical imaging case volume changes over time, the total of each year was calculated, including the minimum, maximum, mean, standard deviation and 95% confident interval range. The ANOVA test was applied to determine differences in the total medical case volume conducted between 2019 to 2022. To compare the means of total medical imaging case volumes conducted in 2020, 2021 and 2022 (cohort groups) against the means of the volume in 2019 (control group), Dunnett’s multiple comparisons’ test was used to establish confidence intervals for differences between the control and cohort groups. Data analysis was performed using GraphPad Prism for MacOS Version 10, and p values < 0.05 was considered statistically significant.

Results

Total medical imaging case volume

Between 2019 and 2022, a total of 597,645 medical imaging procedures were performed (Table 1). The lowest monthly volume (n = 5,325) occurred in April 2020, while the highest (n = 15,997) was recorded in June 2021 (Fig. 1). In 2019, the annual total was 138,326 procedures. The number of procedures decreased by 3.81% in 2020, resulting in a total of 133,054 procedures. However, there was an increase in the subsequent years, with 158,717 procedures in 2021 and 167,548 in 2022. An analysis of variance (ANOVA) test indicated a significant difference in the annual volumes between 2019 and 2022 (F = 6.69, P = 0.0008) (Table 2, Fig. 1). Multiple comparisons using Dunnett’s test revealed that the 2019 volume (11,527 ±  1,807) was significantly lower than 2022 (13,365 ± 940.6) (p = 0.006). However, no significant differences were found between 2019 (11,527 ± 1,807) and 2020 (11,088 ±  2,672) (p = 0.88), and between 2019 (11,527 ± 1,807) and 2021 (13,226 ± 1,443) (P = 0.07) (Table 2, Fig. 1).

Table 1 Descriptive statistics of medical imaging case volume from 2019 to 2022.

Modality	Descriptive statistics	
	Descriptive test	Control - 2019	Cohort - 2020	Cohort - 2021	Cohort - 2022	
All	Total (Min–Max)	138,326 (8,809–14,494)	133,054 (5,325–13,414)	158,717 (11,030–15,997)	167,548 (12,004–14,954)	
Mean (Std. Deviation)	11,527 (1,807)	11,088 (2,672)	13,226 (1,443)	13,962 (940.6)	
95% CI (Lower–Upper)	(10,379–12,675)	(9,390–12,786)	(12,310–14,143)	(13,365–14,560)	
BMD\CR	Total (Min–Max)	1,183 (67–138)	716 (1–131)	708 (0–105)	945 (16–117)	
Mean (Std. Deviation)	98.58 (23.98)	59.67 (38.04)	59 (36.7)	78.75 (26.92)	
95% CI (Lower–Upper)	(83.35–113.8)	(35.49–83.84)	(35.68–82.32)	(61.65–95.85)	
CT	Total (Min–Max)	19,055 (1,420–1,768)	20,287 (905–2,011)	24,537 (1,780–2,445)	25,797 (1,931–2,385)	
Mean (Std. Deviation)	1,588 (109.2)	1,691 (353.7)	2,045 (201)	2,150 (130)	
95% CI (Lower–Upper)	(1,519–1,657)	(1,466–1,915)	(1,917–2,172)	(2,067–2,232)	
DX	Total (Min–Max)	95,719 (5,725–10,566)	90,187 (3,951–8,923)	105,419 (7,083–10,797)	112,898 (8,482–10,190)	
Mean (Std. Deviation)	7,977 (1,444)	7,516 (1,612)	8,785 (1,005)	9,408 (578.6)	
95% CI (Lower–Upper)	(7,059–8,894)	(6,491–8,540)	(8,146–9,423)	(9,041–9,776)	
MG	Total (Min–Max)	1,151 (31–191)	998 (5–129)	1,364 (59–142)	1,311 (44–148)	
Mean (Std. Deviation)	95.92 (46.52)	83.17 (41.63)	113.7 (27.92)	109.3 (32.05)	
95% CI (Lower–Upper)	(66.36–125.5)	(56.72–109.6)	(95.93–131.4)	(88.89–129.6)	
MR	Total (Min–Max)	7,651 (372–801)	7,309 (85–1,013)	9,829 (572–1,010)	10,116 (506–1,006)	
Mean (Std. Deviation)	637.6 (136.5)	609.1 (319)	819.1 (157.3)	843 (139.8)	
95% CI (Lower–Upper)	(550.9–724.3)	(406.4–811.8)	(719.1–919)	(754.2–931.8)	
NM	Total (Min–Max)	306 (1–51)	347 (2–48)	383 (18–48)	436 (1–67)	
Mean (Std. Deviation)	25.5 (16.99)	28.92 (14.11)	31.92 (8.03)	36.33 (15.98)	
95% CI (Lower–Upper)	(14.71–36.29)	(19.95–37.88)	(26.81–37.02)	(26.18–46.49)	
NM/CT	Total (Min–Max)	171 (0–32)	214 (0–31)	304 (16–39)	340 (4–52)	
Mean (Std. Deviation)	14.25 (10.37)	17.83 (10.51)	25.33 (7.71)	28.33 (14.85)	
95% CI (Lower–Upper)	(7.66–20.84)	(11.15–24.51)	(20.43–30.23)	(18.9–37.77)	
RF	Total (Min–Max)	170 (6–34)	142 (1–34)	239 (9–70)	157 (2–20)	
Mean (Std. Deviation)	14.17 (8.2)	12.91 (9.22)	19.92 (16.45)	13.08 (5.50)	
95% CI (Lower–Upper)	(8.95–19.38)	(6.71–19.11)	(9.46–30.37)	(9.58–16.58)	
US	Total (Min–Max)	12,182 (765–1,386)	12,059 (323–1,330)	15,179 (857–1,511)	14,749 (858–1,510)	
Mean (Std. Deviation)	1,015 (221.9)	1,005 (362.5)	1,265 (214.6)	1,229 (176.5)	
95% CI (Lower–Upper)	(874.2–1,156)	(774.6–1,235)	(1,129–1,401)	(1,117–1,341)	
XA	Total (Min–Max)	738 (40–92)	795 (35–83)	755 (51–80)	799 (53–89)	
Mean (Std. Deviation)	61.5 (14.95)	66.25 (13.53)	62.92 (9.13)	66.58 (10.93)	
95% CI (Lower–Upper)	(52–71)	(57.66–74.84)	(57.12–68.72)	(59.64–73.53)	

Figure 1 The change in imaging case volume of all modalities over time from January 2019 to December 2022.

Table 2 ANOVA and multiple comparison tests of medical imaging case volume from 2019 to 2022.

Modality	ANOVA	Multiple comparisons	
	F	P value	Dunnett’s multiple comparisons test	Mean Diff.	95.00% CI of diff.	P Value	
			Cohort - 2020 vs. Control - 2019	−439.3	−2,255 to 1,377	0.88	
All	6.698	0.0008***	Cohort - 2021 vs. Control - 2019	1,699	−116.7 to 3,515	0.07	
			Cohort - 2022 vs. Control - 2019	2,435	619.2 to 4,251	0.006**	
			Cohort - 2020 vs. Control - 2019	−38.92	−70.69 to −7.140	0.01*	
BMD\CR	4.132	0.01*	Cohort - 2021 vs. Control - 2019	−39.58	−71.36 to −7.807	0.01*	
			Cohort - 2022 vs. Control - 2019	−19.83	−51.61 to 11.94	0.30	
			Cohort - 2020 vs. Control - 2019	102.7	−116.3 to 321.6	0.53	
CT	18.16	<0.0001****	Cohort - 2021 vs. Control - 2019	456.8	237.9 to 675.8	<0.0001****	
			Cohort - 2022 vs. Control - 2019	561.8	342.9 to 780.8	<0.0001****	
			Cohort - 2020 vs. Control - 2019	−461	−1680 to 758.2	0.68	
DX	5.639	0.002**	Cohort - 2021 vs. Control - 2019	808.3	−410.9 to 2,028	0.26	
			Cohort - 2022 vs. Control - 2019	1,432	212.3 to 2,651	0.01*	
			Cohort - 2020 vs. Control - 2019	−12.75	−50.26 to 24.76	0.74	
MG	1.603	0.20	Cohort - 2021 vs. Control - 2019	17.75	−19.76 to 55.26	0.52	
			Cohort - 2022 vs. Control - 2019	13.33	−24.17 to 50.84	0.72	
			Cohort - 2020 vs. Control - 2019	−28.5	−230.0 to 173.0	0.97	
MR	4.259	0.01*	Cohort - 2021 vs. Control - 2019	181.5	−20.04 to 383.0	0.08	
			Cohort - 2022 vs. Control - 2019	205.4	3.872 to 407.0	0.04*	
			Cohort - 2020 vs. Control - 2019	3.417	−10.70 to 17.53	0.88	
NM	1.257	0.30	Cohort - 2021 vs. Control - 2019	6.417	−7.697 to 20.53	0.55	
			Cohort - 2022 vs. Control - 2019	10.83	−3.280 to 24.95	0.16	
			Cohort - 2020 vs. Control - 2019	3.583	−7.499 to 14.67	0.77	
NM/CT	4.093	0.01*	Cohort - 2021 vs. Control - 2019	11.08	0.001375 to 22.17	0.04*	
			Cohort - 2022 vs. Control - 2019	14.08	3.001 to 25.17	0.0095**	
			Cohort - 2020 vs. Control - 2019	−1.258	−12.11 to 9.600	0.98	
RF	1.142	0.34	Cohort - 2021 vs. Control - 2019	5.75	−4.869 to 16.37	0.41	
			Cohort - 2022 vs. Control - 2019	−1.083	−11.70 to 9.535	0.98	
			Cohort - 2020 vs. Control - 2019	−10.25	−262.4 to 241.9	0.99	
US	3.527	0.02*	Cohort - 2021 vs. Control - 2019	249.8	−2.441 to 501.9	0.052	
			Cohort - 2022 vs. Control - 2019	213.9	−38.27 to 466.1	0.11	
			Cohort - 2020 vs. Control - 2019	4.75	−7.508 to 17.01	0.66	
XA	0.493	0.68	Cohort - 2021 vs. Control - 2019	1.417	−10.84 to 13.67	0.98	
			Cohort - 2022 vs. Control - 2019	5.083	−7.174 to 17.34	0.62	
Notes.

Significant results were considered as P values < 0.05 that were flagged as followings: P values < 0.05 are shown with one asterisk (*), P values < 0.01 are shown with two asterisks (**), P values < 0.001 are shown with three asterisks (***), P values < 0.0001 are shown with four asterisks (****).

Medical imaging modalities case volume

Specifically, the bone mineral density/computed radiography (BMD/CR) modality experienced a considerable decrease in procedures in 2020 (59.67 ± 38.04) and 2021 (59.00 ± 36.70) compared to 2019 (98.58 ± 23.98) (Fig. 2). Conversely, the computed tomography (CT) modality demonstrated a significant increase in 2021 (2,045 ± 201) and 2022 (2,150 ± 130) compared to the pre-pandemic year 2019 (1,588 ± 109.2) (Fig. 2). The digital X-ray modality (DX) showed the highest volume of procedures overall between 2019 and 2022 that was constituting 67.63% of the overall medical imaging procedures (Fig. 2). Meanwhile, magnetic resonance imaging (MR) and ultrasound (US) modalities experienced a slight drop in the number of procedures in 2020–4.47% for MR and 1.00% for US which subsequently recovered by 22.15% and 19.74% in 2021, and 24.36% and 17.40% in 2022, respectively, compared to 2019 (Fig. 2).

ANOVA tests indicated a significant difference in the number of BMD/CR, CT, DX, MR, NM/CT and US procedures performed between 2019 and 2022 (F = 4.13, P = 0.01), (F = 18.16, P <  0.0001), (F = 5.63, p = 0.002), (F = 4.25, p = 0.01), (F = 4.09, p = 0.01) and (F = 3.52, P = 0.02), respectively (Fig. 2, Table 1). On the other hand, no significant differences were found in the number of MG, NM, RF and XA procedures performed between 2019 and 2022 (F = 1.60, P = 0.2), (F = 1.25, p = 0.3), (F = 1.14, p = 0.34) and (F = 0.59, P = 0.68), respectively (Fig. 2, Table 1).

Multiple comparisons using Dunnett’s tests revealed a significant difference (P = 0.01) of the number of BMD/CR procedures performed between 2019 (98.58 ± 23.98) and 2020 (59.67 ± 38.04). Also, significant differences were found between the number of BMD/CR (P = 0.01), CT (p < 0.0001) and NM/CT (p = 0.04) procedures performed between 2019 (98.58 ± 23.98), (1,588 ± 109.2) and (14.25 ± 10.37), respectively, and 2021 (59.00 ± 36.70), (2,045 ± 201) and (25.33 ± 7.715), respectively. In addition, significant differences were found between the number of CT (p < 0.0001), DX (P = 0.01), MR (P = 0.04) and NM/CT (P = 0.009) procedures performed between 2019 (1,588 ± 109.2), (7,977 ± 1444), (636.6 ± 136) and (14.25 ± 10.37), respectively, and 2022 (2,150 ± 130), (9,408 ± 578.6), (843 ± 139.8) and (28.33 ± 14.85), respectively (Fig. 2, Table 2).

Discussion

The investigation into the extended impact of the COVID-19 pandemic on medical imaging case volumes has unveiled insights that are unprecedented in existing literature. This study, addressing a previously unexplored aspect of the pandemic’s aftermath, aligns with its initial objective of assessing long-term changes in various medical imaging modalities from 2019 to 2022. The findings reveal significant fluctuations in imaging case volumes, underscoring the pandemic’s profound and multifaceted impact on medical imaging practices.

Figure 2 The monthly change of imaging case volume for each medical imaging modality from January 2019 to December 2022.

The results of this study indicate a modest decrease in the total number of medical imaging procedures performed in 2020 compared to 2019. This aligns with a study carried out in the Aseer region of Saudi Arabia, which indicated a decrease in overall imaging volume in 2020 when compared to 2019 (Alelyani et al., 2021). This reduction is likely attributable to the healthcare delivery restrictions implemented during the first wave of the COVID-19 pandemic. These measures, including the cancellation or postponement of a large number of elective surgeries and non-urgent medical services, were essential to preserve hospital resources, personal protective equipment, and to minimize infection risk to patients and healthcare workers (COVIDSurg Collaborative, 2020; Sreedharan et al., 2021). The current study highlighted a return to pre-pandemic levels in the following years, noting an uptick in medical imaging procedures in 2021 and 2022. This rise may be attributed to the introduction of new practices, including pre-appointment phone screenings for outpatients, the reorganization of departments and workflows, and the implementation of guidelines created by the National Institute for Health and Care Excellence and specialist societies (COVID-19 rapid guideline: arranging planned care in hospitals and diagnostic services, 2020), which have supported the safe resumption of diagnostic services (Peprah et al., 2021). Additionally, the incorporation of artificial intelligence (AI) in the radiographic detection of COVID-19 has proven beneficial, enhancing imaging-based diagnosis with improvements in both diagnostic accuracy and speed (DeGrave, Janizek & Lee, 2021; Kriza et al., 2021). In terms of streamlined resource allocation, it has been determined that multimodal imaging can be essential for diagnosing and categorizing patients during a potential second wave of COVID-19 (Varadarajan et al., 2021). Furthermore, teleradiology has demonstrated a beneficial impact during COVID-19 by reducing the risk of infection and alleviating workload (Hammond & Gunderman, 2021; Al-Dahery et al., 2023).

The COVID-19 pandemic had its most pronounced impact on the BMD/CR modality, with a significant decrease in the number of procedures in 2020 and 2021 compared to 2019. Nonetheless, this trend saw a robust recovery in 2022, with a 25% increase compared to the previous year. This surge could be attributed to the easing of COVID-19 restrictions, which enabled patients to schedule appointments for BMD/CR examinations. BMD/CR, also known as dual-energy X-ray absorptiometry (DEXA), is crucial for calculating bone density. This is essential for diagnosing osteoporosis, assessing fracture risk, and monitoring osteoporosis treatment effectiveness. Recognized as the gold standard and the most accurate test for osteoporosis and fracture risk prediction (Pisani et al., 2013), the reduction in BMD screening could potentially delay early osteoporosis diagnosis. This underscores the need for careful consideration in managing BMD screening, especially in times of healthcare crises, to prevent delayed diagnosis and treatment of critical conditions like osteoporosis.

To address the decline in BMD/CR screenings during the COVID-19 pandemic, implementing extending working hours to accommodate additional appointments, along with measures to limit patient numbers in the department at any given time, are viable and effective strategies. These approaches, while ensuring adherence to safety protocols, could effectively manage the backlog of cases and reduce wait times for critical diagnostic procedures. Additionally, there is a pressing need to carefully consider the balance between the risk of COVID-19 transmission and the importance of conducting these tests. A critical evaluation is necessary to determine whether the benefits of BMD/CR testing outweigh the potential risks of COVID-19 exposure, particularly considering the urgency of osteoporosis diagnosis and treatment against the health risks posed by the pandemic for vulnerable populations. Implementing such strategies and assessments could facilitate the continuation of essential medical imaging services, ensuring that crucial diagnoses are not delayed.

In the current study, CT modality exhibited a notable increase in the number of procedures performed in 2021 and 2022 compared to the pre-pandemic year 2019. This trend aligns with findings from other research (Fleckenstein et al., 2022; Bonacossa de Almeida et al., 2023), which also reported an increase in the number of CT procedure in the post-pandemic period. The Saudi Food & Drug Authority reported a 143% decrease in CT requests in public hospitals and a 40% increase in private hospitals in 2020 compared to 2019 (The Saudi Food & Drug Authority (SFDA), 2021). Similarly, DX modality recorded the highest volume of procedures performed between 2019 and 2022. The broad application of DX in diagnosing a variety of pathological disorders may contribute to this trend. For instance, in Brazil, the number of DX procedures in outpatient setting was declined by 21.1% in 2020 and by 12.7% in 2021 compared to 2019, and in inpatient setting, the trend was declined by 3.2% in 2020 and then increased by 7.0% in 2021 (Bonacossa de Almeida et al., 2023). In 2020, the Saudi Food & Drug Authority reported a 43% drop in DX requests at public hospitals and a 4% decline in private hospitals compared to the previous year, 2019 (The Saudi Food & Drug Authority (SFDA), 2021). In contrast, Germany observed 30% increase in DX procedures during 2020 and 2021 compared to 2019 (Fleckenstein et al., 2022). Furthermore, a substantial increase in mobile X-ray usage was noted following the pandemic in Germany (Yeung et al., 2022). The escalating trends in CT and DX procedures observed in this study are likely attributable to the heightened need for these diagnostic modalities in identifying and managing COVID-19 cases. Notably, most of these procedures were chest CT scans and X-ray examinations, which have been pivotal in detecting COVID-19-related pulmonary abnormalities (Costa et al., 2022).

Throughout the pandemic, there was a noticeable reduction in the number of MR and US procedures, with a drop of 4.47% for MR and 1.00% for US in 2020 compared to 2019. However, these modalities showed a strong recovery in the following years, with increases of 22.15% and 19.74% in 2021, and 24.36% and 17.40% in 2022, respectively. This trend aligns with the findings of Bonacossa de Almeida et al. (2023), who reported a similar decline in MR and US procedures by 1.2% and 6.2%, respectively in 2020, followed by increases of 8.9% and 2.4% in 2021 compared to 2019. Furthermore, Alelyani et al. (2021) noted a substantial decrease in MR and US imaging volume in 2020 compared to 2019. In addition, a significant proportion of MRI and US scans might be categorized as non-urgent, which likely contributed to the slight drop in the volume of these procedures. This reduction can be attributed to the prioritization of urgent and COVID-19 related cases, leading to the postponement or rescheduling of routine and non-critical imaging.

Notably, the imaging case volumes of certain modalities, including MG, NM, RF and XA, were not significantly influenced by COVID-19 pandemic. In the current study, no significant difference was observed in the number of MG between 2019 and 2022. This aligns with Alelyani et al. (2021), who reported no significant difference of MG procedures in 2020 when compared to 2019. However, this finding contrasts with global reports of substantial reductions in breast cancer screening due to the pandemic. Cairns et al. (2022) found a significant decrease in screening mammograms in 2020 compared to 2019, although there was no significant difference in mammogram diagnostics and breast cancer operations between the periods of prior to and during the pandemic. In agreement with the finding of our study, NM services were among the least impacted in terms of imaging case volume, compared to other modalities during the pandemic (Sreedharan et al., 2021). It is worth noting that NM examinations are heavily reliant on the availability of radioactive materials, which are necessary for these procedures. Even before the COVID-19 pandemic, cancellations and rescheduling of NM exams were frequently expected due to disruptions in the supply chain for these essential materials. Furthermore, an assessment of the pandemic’s effect on cardiac care revealed significant improvements in prescription for secondary prevention and referrals to cardiac rehabilitation services during the COVID-19 pandemic compared to pre-pandemic period (Hussain et al., 2021).

The primary limitation of this study lies in its single-site scope. Other hospitals in the region were unable to provide the requested data because of the incomplete information obtained from the Integrated Radiology Information System-Picture Archive and Communication System (RIS-PACS). As only one site was included, the findings may not comprehensively represent the impact of COVID-19 on the medical imaging case volumes across the region. To enhance the generalizability of the results, future studies should consider including multiple sites. Additionally, the retrospective design of our study imposed certain constraints, limiting the analysis to modality types, and precluding more detailed variables such as specific departments, patient types (inpatient, outpatient and emergency patient), workforce status (number of radiographers performing the procedures and the number of radiologists reporting during the pandemic) and varied procedure categories (brain, abdomen, and chest imaging). For a more nuanced understanding, it is highly recommended that future research adopts a prospective study design. This approach would allow for the inclusion of a broader range of variables, especially those with specific exposure data, leading to a more comprehensive and detailed analysis.

Conclusions

The COVID-19 pandemic initially led to a notable reduction in medical imaging procedures in 2020, with this decline being particularly pronounced during the early waves of the pandemic. However, the study observed a gradual recovery in imaging volumes in 2021 and 2022. These findings hold significant implications for public health, especially in understanding the extended effects of the COVID-19 pandemic on medical imaging case volumes. Insights from this study could inform the development of proactive recovery strategies and plans to better prepare for and mitigate the impacts of future pandemics on essential medical services. Such strategies might include enhancing resource allocation, improving crisis response mechanisms in medical imaging departments, and ensuring the continuity of critical diagnostic services even under challenging circumstances.

Supplemental Information

Supplemental Information 1 Raw data of the monthly totals of all medical imaging procedures conducted from 2019 to 2022

The authors extend their heartfelt thanks to the General Directorate of Health Affairs in Almadinah Almunawarah, Saudi Arabia, for their collaboration in the data collection process. This article was refined with the help of ChatGPT (ChatOn-AI Chatbot Assistance), which contributed to enhancing the clarity and coherence of the content.

Additional Information and Declarations

Competing Interests

Author Contributions

Human Ethics

Data Availability

The authors declare there are no competing interests.

Fahad H. Alhazmi conceived and designed the experiments, analyzed the data, prepared figures and/or tables, authored or reviewed drafts of the article, and approved the final draft.

Faisal A. Alrehily conceived and designed the experiments, authored or reviewed drafts of the article, and approved the final draft.

Walaa Alsharif conceived and designed the experiments, authored or reviewed drafts of the article, and approved the final draft.

Moawia Gameraddin performed the experiments, analyzed the data, prepared figures and/or tables, and approved the final draft.

Kamal D. Alsultan performed the experiments, analyzed the data, prepared figures and/or tables, and approved the final draft.

Hassan Ibrahim Alsaedi performed the experiments, analyzed the data, prepared figures and/or tables, and approved the final draft.

Khalid M. Aloufi performed the experiments, analyzed the data, prepared figures and/or tables, and approved the final draft.

Sultan Abdulwadoud Alshoabi performed the experiments, analyzed the data, prepared figures and/or tables, and approved the final draft.

Osamah M. Abdulaal conceived and designed the experiments, performed the experiments, analyzed the data, prepared figures and/or tables, and approved the final draft.

Abdulaziz A. Qurashi conceived and designed the experiments, authored or reviewed drafts of the article, and approved the final draft.

The following information was supplied relating to ethical approvals (i.e., approving body and any reference numbers):

This study has obtained ethical approval from the Institutional Review Board of the General Directorate of Health Affairs in Medina, Saudi Arabia (Ref. 031-22).

The following information was supplied regarding data availability:

The raw data are available in the Supplementary File.

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
