# Peer review of "The extended impact of the COVID-19 pandemic on medical imaging case volumes: a retrospective study"

_PeerJ, doi:10.7717/peerj.18987_

## Round 0.1 · original submission · Major Revisions

Please respond to all the questions and issues from both reviewers.

Reviewer 1 ·

Basic reporting

The manuscript is clear, well-structured, professionally written, and includes sufficient background and relevant literature. Figures, tables, and raw data are appropriately presented, and the study is self-contained with results that align well with the hypotheses.

Experimental design

The manuscript fits well within the journal’s scope, presenting original, primary research. The research question is clearly defined, relevant, and addresses a meaningful knowledge gap. The study follows rigorous technical and ethical standards, with methods detailed enough to enable replication.

Validity of the findings

The manuscript does not assess impact or novelty but encourages meaningful replication, clearly stating its rationale and benefits to the literature. All underlying data are robust, statistically sound, and well-controlled. Conclusions are clearly articulated, directly linked to the research question, and confined to the supporting results.

Additional comments

I appreciate the chance to review.
-Almost three years after the COVID-19 outbreak, the paper remains relevant. In radiology, writers should highlight long-term flexibility and resilience. How imaging departments have altered workflows, integrated AI, and simplified resource allocation will influence healthcare disruption management. Pandemic data-based predictive modeling and patient-centered developments like tele-imaging and outpatient treatments can provide forward-looking perspectives on imaging demand.
-Please describe the COVID-19 pandemic from January 2019 to December 2022, which ended in 2022.
-Insufficient methodology details in the abstract.
-The total amount of images? The abstract does not say if the authors focused on covid-19 or all cases.
-The investigation's goals are unclear.
-The single-institution study did not investigate work load before, during, and after COVID-19. Why is this pandaic center or general hospital the only large one in the region?
-The long opening has many mismatched sentences.
-Why did writers incorporate non-COVID imaging?
- Unjustified changes in imaging modalities, such as BMD rise in 2019, were seen. Hospitals nearly closed due to COVID-19.
-The results section needs subheadings for clarity.
-The publication mentions breast cancer as one of the most common tumors worldwide and being identified largely by mammography (Miller, 2001). What does this information add to the discussion?
No comparisons have been made with relevant studies in the country or region.

Reviewer 2 ·

Basic reporting

Sufficient literature references from various sources. Figures and tables are appropriate. Although would like to suggest that the authors keep the decimal point to 1 instead of 2 which is currently used. 1 decimal point would be more appropriate and easier to read for the various values in the text and tables; for your consideration.

Experimental design

1 query: did you extract data from PACS or from RIS? PACS primarily store and digitally transmit electronic images and clinically-relevant reports but most medical imaging associated data should be available in RIS? Perhaps would also be appropriate to name the vendor for transparency in your approach? And likewise, is there no possibility in identifying the category for the patient, e.g. inpatient, out patient or emergency patient, as this will definitely value add to your findings and place you in a better position to " identify best practices and inform future healthcare policy and preparedness strategies, with the goal of maintaining essential medical imaging services during health crises and ensuring the resilience and responsiveness of healthcare delivery systems".

Validity of the findings

No further comment. Just a follow-up to my prior point on "seeks to identify best practices and inform future healthcare policy and preparedness strategies, with the goal of maintaining essential medical imaging services during health crises and ensuring the resilience and responsiveness of healthcare delivery systems". I am not fully convinced that the data available would aid in addressing the aforementioned; though the trend is well illustrated.

Additional comments

Perhaps without reinventing the wheel, is there any possibility of further dictomising the data for patient category or group or even age etc for more meaningful interpretation that will help to identify best practices and inform future healthcare policy? E.g. workforce such as number of radiographers performing the procedures and the number of radiologists reporting during the upwards trend or downwards trend? Reallocation of workforce etc? At the moment, the aim to elucidate the volume trend is well documented but beyond that there may still room to expand on the content, drawing from the data available.

---

## Round 0.2 · Minor Revisions

Please respond to the questions from the reviewer

Reviewer 1 ·

Basic reporting

CHeck below

Experimental design

CHeck below

Validity of the findings

CHeck below

Additional comments

-The single-institution study did not investigate work load before, during, and after COVID-19. Why is this pandemic center or general hospital the only large one in the region?
The author didn’t answer my questions. To clarify, Why did you select this hospital from all the others in the region? Is it a COVID center? How does COVID affect this hospital and other regional or city hospitals? What is the difference between the workload?

-No comparisons have been made with relevant studies in the country or region.
In response to my comment, you used only one reference, which is not accepted I am familiar with many papers in your country and GCC countries with similar topics

Reviewer 2 ·

Basic reporting

No comment

Experimental design

No comment

Validity of the findings

No comment

Additional comments

No comment

---

## Round 0.3 · accepted · Accept

I think the current version can be Accepted